# The Potential Role of 3D In Vitro Acute Myeloid Leukemia Culture Models in Understanding Drug Resistance in Leukemia Stem Cells

**DOI:** 10.3390/cancers14215252

**Published:** 2022-10-26

**Authors:** Basil Al-Kaabneh, Benjamin Frisch, Omar S. Aljitawi

**Affiliations:** 1Hematology/Oncology Division, University of Rochester Medical Center, Rochester, NY 14642, USA; 2Departments of Pathology and Biomedical Engineering, University of Rochester Medical Center, Rochester, NY 14642, USA

**Keywords:** acute myeloid leukemia (AML), allogeneic hematopoietic stem cell transplant (Allo-HSCT), leukemia stem cells (LSCs), minimum residual disease (MRD), bone marrow (BM), extra cellular matrix (ECM), mesenchymal stem cells (MSCs), decellularized Wharton jelly matrix (DWJM), hematopoietic stem/progenitor cells (HSPCs), myelodysplastic syndrome (MDS)

## Abstract

**Simple Summary:**

Despite advancements in the treatment of hematological malignancies, disease relapse is still a major concern in the management of acute myeloid leukemia (AML). It is believed that chemotherapeutic resistance in leukemic stem cells is the driving factor behind disease relapse. The role of in vitro 3D models in studying the mechanisms of chemotherapeutic drug resistance in leukemic stem cells is not well-researched. Herein, we review the recent advancements in using 3D in vitro models in studying AML. In addition, we describe their role as potential platforms for understanding mechanisms of resistance and relapse in AML and in identifying and testing therapeutic modalities.

**Abstract:**

The complexity of the bone marrow (BM) microenvironment makes studying hematological malignancies in vitro a challenging task. Three-dimensional cell cultures are being actively studied, particularly due to their ability to serve as a bridge of the gap between 2D cultures and animal models. The role of 3D in vitro models in studying the mechanisms of chemotherapeutic resistance and leukemia stem cells (LSCs) in acute myeloid leukemia (AML) is not well-reviewed. We present an overview of 3D cell models used for studying AML, emphasizing the recent advancements in microenvironment modeling, chemotherapy testing, and resistance.

## 1. Introduction

Despite advancements in understanding acute myeloid leukemia (AML) pathogenesis and its therapy, most patients with AML continue to have grim long-term outcomes. Standard conventional chemotherapy induces remission in 50% to 75% of adult patients with AML, but unfortunately only 20% to 30% of these patients enjoy long-term leukemia-free survival [1]. In addition, remissions in relapsed and refractory patients are not durable and most AML patients die of their disease [2]. Disease relapse after an apparent response is attributed to the growth of leukemic cells that persist despite chemotherapy toxicity, or immune response in the case of allogeneic hematopoietic stem cell transplantation (Allo-HSCT); these cells are clinically referred to as minimal residual disease (MRD) [3], with studies confirming the relationship between MRD detection after chemotherapy treatment and the risk of relapse [4]. Studies also show that MRD negativity is associated with improved leukemia survival and disease control [5].

Biologically, MRD might include different cellular components, including leukemic stem cells (LSCs), which are believed to contribute to disease initiation at the time of relapse [6]. Using primitive stem cell markers like CD34, CD117, or CD133, a study looking into different immunophenotypic markers for MRD assessment reliably predicted survival [7]. Other studies found that high stem cell frequency at diagnosis predicted high MRD and poor survival [8]. These observations support that LSCs are an important component of AML MRD. To understand the leukemia stem cell concept, it is important to recognize that AML is a heterogeneous disease, with different populations of leukemic cells. Within the AML population, only 1% of leukemic cells are clonogenic progenitors (AML-colony-forming units (AML-CFU)) based on in vitro studies [9], which is the notion that led to the discovery of the so-called leukemic stem cells (LSCs). LSCs are considered a minute portion of the leukemic blast population (1 per 10^6^ leukemic blasts), but they are capable of propagating AML in appropriate xenograft models [10]. Currently, AML populations are best viewed as a hierarchy that originates from leukemia-initiating cells or LSCs, which in turn produce AML colony-forming units and leukemic blasts [10]. These LSCs are equipped with several chemotherapy- and immune therapy-protective mechanisms. Accordingly, to prevent relapse a successful treatment strategy should target the LSCs directly or target pathways important for LSCs survival.

To understand LSCs survival mechanisms, it is crucial to understand how leukemia cells interact with different elements of the 3D bone marrow microenvironment [11]. Such need has fueled the interest in developing 3D in vitro AML culture models in the last decade. These models will certainly help us understand LSCs’ interactions with their microenvironment, and how these interactions support LSCs survival and resistance to chemotherapy (and immune response in the case of Allo-HSCT). These models will be very helpful in developing and testing interventions that target LSCs interactions with the microenvironment [12]. In trying to mimic living 3D microenvironments, in vitro models have been incorporating the three main components of a living microenvironment to variable degrees. These main components are cells, matrix, and soluble factors [13]. Three-dimensional interactions between cells, between cells and matrix, and between cells and soluble factors ultimately influence cell behavior, including AML and LSCs.

## 2. Materials and Methods

In this minireview, the literature review was performed by using the online search engine PubMed (https://pubmed.ncbi.nlm.nih.gov/, accessed on 1 April 2022) to look for articles containing the specific keywords mentioned below, in the period between 2012 and 2022. We also examined the references section of initially reviewed articles for other relevant articles. Keywords used for the retrieval of articles included “three-dimensional”, “in vitro”, “acute myeloid leukemia”, and “drug resistance”.

## 3. Overview of In Vitro 3D AML Models

Three-dimensional in vitro models for hematological malignancies can be categorized as scaffold-based or scaffold-free. Scaffold-free models enable cells to morph into spheroids and grow in the absence of an anchor. In the absence of a scaffold, the accurate biomimicking of cellular in vivo interactions that take place in the BM becomes limited, especially the interactions between malignant cells and the ECM. The significance of scaffolds lies in providing the medium which allows cell–matrix interactions and supports anchorage-related growth of leukemic cells.

In more detail, scaffolds are either synthetic or biologic. Synthetic scaffolds are biocompatible hydrogels or polymers; the benefit of these scaffolds is that they are also flexible, widely available, consistent, and less expensive. In the context of biological scaffolds, they are composed of natural products abundant in the ECM of the BM, therefore allowing more accurate recapitulation of the BM microenvironment. In addition, they are accessible, and they do not require specialized equipment or extensive knowledge of all components of the niche.

Previously, 2D models were used to study leukemic interactions in the BM niche. However, these models were limited by only providing cell–cell interactions, and therefore a less-reliable platform for studying the BM microenvironment. On the contrary, the structure of 3D models includes the main three components of the BM, which are cells, matrix, and soluble factors, hence mirroring the actual living BM microenvironment. In this review, we will discuss the current 3D AML models and describe the most recent advancements in in vitro 3D models.

### 3.1. Synthetic Scaffold-Based 3D Models (Table 1)

The bone marrow microenvironment niche has been studied extensively in hematological malignancies to gain a better understanding of the mechanisms of resistance to chemotherapy. In one of our previous synthetic scaffold-based 3D models, we cultured AML cells with mesenchymal stem cells (MSCs) and demonstrated superior results in terms of chemoresistance to daunorubicin and increased N-cadherin expression when compared to 2D models or suspension conditions [14], which also supports the role of adhesion molecules in inducing chemoresistance in AML cells [14]. Bray et al. designed a 3D triculture model to evaluate cell-to-cell interactions in vitro and to further study the vascular niche [15]. Results from this study supported that 3D models are more accurate in assessing chemotherapeutic resistance than 2D models. Results also showed an increased mobilization of leukemic cells when AMD3100 (a CXCR4 antagonist) was administered, thus enabling the visualization and analysis of AML cells’ behavior when they interact with the vascular niches.

To study the cellular response of AML cells under environmental stress, Velliou et al. used a highly porous PU collagen-coated scaffold to evaluate the cellular proliferation and metabolic profile of AML cells in vitro. Results showed that when the supply of oxygen and glucose to K-652 AML cells is limited, cells still maintain their adaptability and survival mechanisms. In contrast to 2D models, 3D models serve as accurate platforms to predict cellular metabolic mechanisms, particularly under environmental stress [16]. While substantial advancements in tissue engineering are occurring, artificial matrices cannot replicate the complex distribution and the variety of signals present in the natural ECM. In addition, the absence of cellular interstitial flow, hematopoietic circulation, and the continuous supply of oxygen and nutrients in these models [14,15,16] compromises their accuracy in mimicking the bone microenvironment.

**Table 1 cancers-14-05252-t001:** Comparison between synthetic scaffold-based 3D models.

Model	Niche Tested	Type of Scaffold	Important Findings
3D stromal-based mode [14]	Stromal niche	Poly glycolic/Poly L-Lactic acid	3D model can better identify chemoresistance and N-cadherin compared to 2D models and cells in suspension
3D Static in vitro models [16]	Stromal niche	PU collagen-coated scaffold	The effect of glucose and oxygen levels on AML cell proliferation and adaptability was more pronounced in 3D models in contrast to 2D models
3D triculture static models [15]	Vascular niche	PEG-heparin hydrogel	Chemoresistance was superior in 3D models when compared to 2D models or cells in suspension

### 3.2. Biologic Scaffold-Based 3D Models (Table 2)

On the other hand, biological scaffolds resemble the tissue of origin, as they contain some constituents of BM ECM. One model utilized decellularized Wharton jelly matrix (DWJM) as a BM ECM mimicking model and found that compared to suspension, their model showed an increase in the markers of leukemic cell quiescence and adhesion-induced chemoresistance compared to suspension [17]. Borella and Da Ros designed an in vitro humanized-3D model which provides a reliable platform for the extensive analysis of cellular behavior in the BM niche. Interestingly, they proved that AML blasts influence MSC activities towards disease progression by connecting to transmembrane proteins, adjusting the transcriptome, and downregulating the immunomodulating potential of AML MSCs collected at the time of diagnosis. Furthermore, they observed similar modifications in healthy mesenchymal h-MSC cells to those of AML MSCs when co-cultured with AML blasts, thereby favoring leukemic progression. Conversely, cells obtained during remission displayed healthy transcriptional and functional features. Nevertheless, the model was also put to use in dual-target therapy testing of a novel drug, lercanidipine (a calcium channel blocker), in combination with targeted chemotherapy agents [18]. Despite using scaffolds that structurally mimic BM tissue, the absence of significant environmental factors, particularly tissue dynamics and BM heterogeneity, restrict their potential to fully recapitulate the kinetics of the BM. Importantly, the major drawback of static models is their limited capability of providing continuous nutrient exchange and waste removal.

**Table 2 cancers-14-05252-t002:** Comparison between biologic scaffold-based 3D models.

Model	Niche Tested	Type of Scaffold	Important Findings
3D biological scaffold-based static model [17]	ECM niche	Decellularized Wharton jelly matrix (DWJM)	3D models demonstrated superior chemoresistance, ALDH + expression, N-cadherin expression
3D- humanized scaffold-based static model [18]	Stromal niche	Hydroxyapatite and collagen	AML blasts alter MSC morphology and on the transcriptome level, 3D models exhibit a platform for testing dual chemotherapy

### 3.3. Dynamic 3D In Vitro Models (Table 3)

Although 3D in vitro models are continuously advancing, their static nature limits our ability to investigate the dynamic interactions occurring in vivo. Accordingly, bioreactors were introduced to provide a dynamic platform and reproduce normal physiological processes similar to those in the tissue microenvironment. Bioreactors are enclosed complex systems which regulate the occurrence of normal biological processes by connecting their sensors to advanced software. The significance of this technology lies in its role in regulating oxygen and nutrient supply, waste removal, and mimicking mechanical and shear stresses occurring in vivo. Additionally, it can be equipped with fully-humanized vasculature and circulating chemokines. As an example, one model has successfully regulated in vivo physiological processes via providing strict control of oxygen gradient, temperature, pH, and accurate spatiotemporal gradient in solid organ tumors [19].

In a 3D bioreactor model, which compared UCSD-AML1 cell line differentiation between one scaffold-free and two different scaffold-based models (osteoblastic and vascular), results confirmed that in the presence of a scaffold in the model, cells were able to expand and differentiate into their mature type. Additionally, when AML cells were added to the cell culture in the model, the resistance to chemotherapy was more pronounced when compared to 2D models. Additionally, leukemic progenitor/stem cells were retained in the scaffold compared to the mature phenotype, which was released in the supernatant [20]. This recapitulates the situation that occurs in vivo where leukemic cells (LCs) reside in the BM in an undifferentiated quiescent state, compared to leukemic cells found in the bloodstream, which exhibit a more differentiated phenotype.

In addition, we note that when AML cells were cultured with components of the osteoblastic niche, there were discrepancies in terms of gene expression and differentiation compared to when cultured with stromal vascular niche components [20].

On the contrary, metabolic profiling of cultured cells would recapitulate the BM microenvironment more accurately and would provide insight into the metabolic behavior of malignant and non-malignant cells in vitro. Despite creating dynamic conditions within models with the use of rotators, stirring techniques, and pumps, there is no direct control over the movements of cells and scaffolds in the culture system, thus resulting in the disruption of cells or the scaffold architecture through collision. Zippel and Raic et al. proposed a dynamic perfused triculture 3D model that mimics both healthy and leukemic BM niches. Investigators introduced a novel method of inducing dynamicism by applying external magnetic fields to the model [21]. Subsequently, they analyzed the differences in the effect of chemotherapeutic agents on AML cells and healthy hematopoietic stem/progenitor cells (HSPCs) simultaneously in terms of cellular viability, proliferation, differentiation, drug resistance, and metabolic profiles. This approach provides a suitable platform to develop and further test new therapeutic agents specific to malignant cells. Its application could potentially be translated into targeted modalities, sparing healthy HSPCs and reducing the burden of hematopoietic toxicity. Although all the aforementioned models provide a high level of complexity, the latter model included cultured stromal cells, which solely represent the stromal niche [21]. Finally, the authors proposed that obtaining cells from supernatant is a reliable analysis method and further protects the scaffold from destruction. To better identify metabolic discrepancies between healthy and AML cells, obtaining individual cells from the scaffold could potentially provide more-accurate results.

Several models have previously been developed to recapitulate the BM microenvironment. One such model from Chou et al. used a micro-physiologic device to demonstrate hematopoietic stem and progenitor cell function in a 3D model with media flow through a vascular channel to provide physiologically relevant nutrient and waste exchange across an endothelial cell barrier [22]. A similar approach was taken by Sharipol et al. that included multiple BM cellular populations, a 3D scaffold, and a vascularized channel with media flow for nutrient and waste exchange. This model used mouse tissues that allowed for competitive repopulation experiments, the gold standard of HSC functional analysis. These studies highlighted the ability to maintain a highly functional HSC population for at least 2 weeks in vitro [23]. These results suggest that similar platforms are highly adaptable to in vitro studies of AML LSCs phenotype and function in future experiments.

**Table 3 cancers-14-05252-t003:** Comparison between dynamic 3D in vitro models.

Model	Niche Tested	Type of Scaffold	Important Findings
Dynamic 3D (bioreactor) humanized model [20]	Vascular niche Stromal niche	Collagen and hydroxyapatite separately	Quiescence and superior chemoresistance in 3D models; highly customizable models which can recapitulate the stromal and vascular niche
Dynamic 3D (magnetic field-based) model [21]	Stromal niche	Magnetic hydrogel	The effect of chemotherapeutic agents on chemoresistance, metabolic profile, proliferation, and differentiation of both HPSC and AML cells

## 4. Discussion

The development of a functional model capable of mimicking the heterogeneity and dynamism of the BM microenvironment with translational potential is still a major challenge. The complexity of the BM microenvironment makes studying hematological malignancies in vitro a challenging task. For a better understanding of AML behavior in the BM niche, we need to take into consideration multiple factors that influence the BM niche. These factors include highly heterogeneous structure, variability in oxygen gradient, discrepancies in BM stiffness, continuous ECM remodeling, and vascularization. Even though the BM microenvironment is dynamic, there is no stringent evidence to favor the use of dynamic models over their static counterparts. This brings us to an unmet need, which is to conduct further experiments under both static and dynamic conditions.

In the context of choosing between scaffolds, Garcia et al. compared phenotypic expression and expansion of AML when cultured in a dynamic 3D stromal vascular model, once with collagen and separately with hydroxyapatite and detected no differences [20]. Thus, the constituents of the scaffold may not significantly influence the results.

Multiple models debated whether extracting MSCs from the BM is more accurate than from umbilical cord tissue. MSCs from both sources can be used in modeling AML, as both can support the BM microenvironment niche. 3D models that utilized umbilical cord tissue cells for culture reported similar results to BMMSCs in terms of chemoresistance, signaling, and proliferative capacity [14,24]. On the other hand, other studies showed that cord tissue MSCs have a higher proliferative capacity and higher nutrient consumption, favoring the use of BM cells instead [25,26].

Despite advancements in cancer biology research, AML relapse is almost inevitable and represents a major hurdle to overcome to improve the quality of care for AML patients. The high relapse rate and drug resistance in AML patients are closely related to the properties of cancer stem cells (CSCs), particularly their quiescent state, complex interplay with the bone marrow (BM) microenvironment, phenotypic plasticity, and their relapse-initiating capacity. Therefore, studying LSCs and their interactions with the BM microenvironment in vitro is an unmet clinical need that could potentially provide invaluable evidence to overcome drug resistance and consequently reduce relapse rates.

Currently, the present models have attempted to accurately mimic the BM microenvironment to obtain strong evidence of the relationship and the interaction between AML cells in the BM niche. Nonetheless, few models focused on studying LSCs and their biological behavior in vitro. LSCs are unique, with characteristics markedly distinguishable from mature AML cells, allowing them to induce relapse. Namely, these cells have the intrinsic ability to differentiate and self-renew [27,28].

In addition, LSCs can avoid being killed by chemotherapeutic agents and radiotherapy by interacting with their microenvironment and surrounding tissues such as adipose tissues [29,30]. Collectively, these properties make LSCs a crucial target for in vitro 3D modeling. Creating an in vitro model would allow us to extensively study these properties and identify important molecular interactions and major clinically relevant targets for future therapeutic implications. The interactions among AML cells, LSCs, and the BM microenvironment play a crucial role in tumor progression and chemotherapeutic resistance. The bone marrow microenvironment is a complex cellular and non-cellular matrix, allowing cell–cell and cell–ECM interactions to reside in the BM occupying the endosteal and sinusoidal niche. Normally these niches regulate the normal functions of hematopoietic stem cells (HSCs). In AML, both endosteal and sinusoidal niches play a key role in LSCs survival, proliferation, differentiation, and chemotherapeutic resistance [28,31,32,33]. These physiological processes are significant in the development of drug resistance, given that stromal cells are capable of inducing drug resistance through either secreting soluble factors or adhesion-mediated mechanisms of resistance. This close relationship between AML and the BM microenvironment fuels the need to better understand this relationship by designing a 3D in vitro model that closely resembles the BM endosteal and sinusoidal niches for future diagnostic purposes and as a target for therapeutic applications.

To successfully reduce the burden of relapse and chemoresistance caused by LSCs, it is crucial to initially identify phenotypically distinctive cell surface markers and the molecular mechanisms that govern their survival. In vivo studies have shown that CD123 and CD96 are highly expressed on the membrane of LSCs, whereas they are weakly expressed on the surface of HSCs and are also involved in the development of leukemia [34,35]. Thus, these results suggest that CD123 and CD96 are not only specific markers for LSCs but also promising targets for therapeutic options. Several studies have shown that LSCs survival is inseparable from its self-renewable features, which are regulated by cellular pathways. During leukemogenesis, these pathways are crucial for AML initiation. Examples of these pathways include B-catenin/WNT, the Hedgehog pathway (Hh), BCL-2, MTOR/PI3K/AKT, and JAK2/STAT3 [36,37,38].

Disease relapse is a major challenge facing clinicians during AML management. Relapsed disease affects almost 10–40% of young patients and a significant percentage of patients 60 years and above [39]. While younger patients with fewer comorbidities often experience complete remission (CR) following a hematopoietic stem cell transplant [40], a large number of patients require a second HSCT due to AML relapse [41].

Consequently, targeted immunotherapies are continuously emerging as a promising management modality for relapsed AML [42]. However, their efficacy and safety profiles are not well-investigated. A group of investigators utilized 3D tissue-engineered bone marrow (3DTEBM) as a platform to test the efficacy of a novel nanoparticle T cell engager (nano-TCE) in killing AML cells in vitro [43]. Thus, the use of 3D models in the era of targeted immunotherapy can serve as a preliminary platform to assess the efficacy and safety of emerging therapeutic agents specific to AML.

Investigators have attempted to conduct clinical trials targeted toward inhibiting pathways related to LSCs initiation, cellular pathways, and survival pathways. In terms of drugs targeting CD123 as an LSCs surface receptor, results have shown insufficient efficacy in treating relapsed AML [44]. On the contrary, many trials have been conducted to study the effect of targeting cellular signaling pathways as a nanotherapeutic agent or in combination with other chemotherapeutic agents. For untreated AML patients or high-risk MDS, the combination of glasedegib (a Hedgehog (Hh) inhibitor) with low-dose cytarabine and daunorubicin was tested in a phase 2 clinical trial, demonstrating good tolerance and inducing investigator-reported complete remission in about 46.4% of patients [45]. In addition, other trials aimed to test the effect of inhibiting other LSCs cellular signaling pathways, such as B-catenin and B-CL2, and those molecules showed a safe and well-tolerated profile [46,47]. In regard to LSCs interaction with the BM microenvironment, treatment of refractory relapsed AML with AMD3100 showed complete remission in 46% of patients [48].

Collectively, the future approach to in vitro modeling of AML should include studying LSCs and their specific properties as they significantly contribute to the resistance to therapy and thus the relapse of AML.

### Future Directions toward Personalized Medicine

The integration of culturing LSCs in a 3D model can potentially take in vitro modeling a step closer to personalized medical plans. By designing 3D in vitro models that closely resemble the BM microenvironment, we can examine these cells and use this information to develop therapeutic plans specific to the phenotype of AML, with fewer side effects and more clinically efficient results. By initially obtaining samples from the patient’s BM and/or peripheral blood and isolating the nuclear cell component, we can subsequently test the isolated cells for their immunophenotype, metabolic patterns, and cellular signaling. This will be followed by culturing patients’ samples in in vitro 3D models that recapitulate BM microenvironmental interactions in order to cultivate and isolate LSCs. During and following culture, these samples could be investigated for specific LSCs cellular signaling pathways, survival mechanisms, and their interaction with the surrounding microenvironment. Identified signaling pathways will serve as potential targets for treatment plans and further provide us with evidence in terms of efficiency and chemoresistance. Furthermore, these newly discovered drugs can also be tested as a mono-therapeutic agent or in conjunction with conventional treatment plans for AML and LSCs cells simultaneously. Collectively, this approach can help us eradicate both AML and LSCs cells (Figure 1).

## 5. Conclusions

Three-dimensional modeling of the BM microenvironment in AML has strengthened our understanding of the intricacies of leukemic cells and important pathogenic mechanisms related to AML survival and adaptation in vivo. Moreover, they have provided us with invaluable techniques to further optimize the designs of potential in vitro models, particularly approaches related to improving the biomimicry of the actual BM microenvironment. With exclusive evidence highlighting the key role of LSCs in disease relapse, there is a clear need to shift our in vitro models toward better identifying LSCs features contributing to disease relapse. While relapse and chemoresistance have always been major hurdles in the management of AML, the literature is lacking models designed specifically to investigate how LSCs induces relapse and chemoresistance. In this mini-review, we believe that potential approaches should be directed toward uncovering LSCs interactions and designing reliable biomimetic platforms for the testing of future therapeutic modalities.

## Figures and Tables

**Figure 1 cancers-14-05252-f001:**
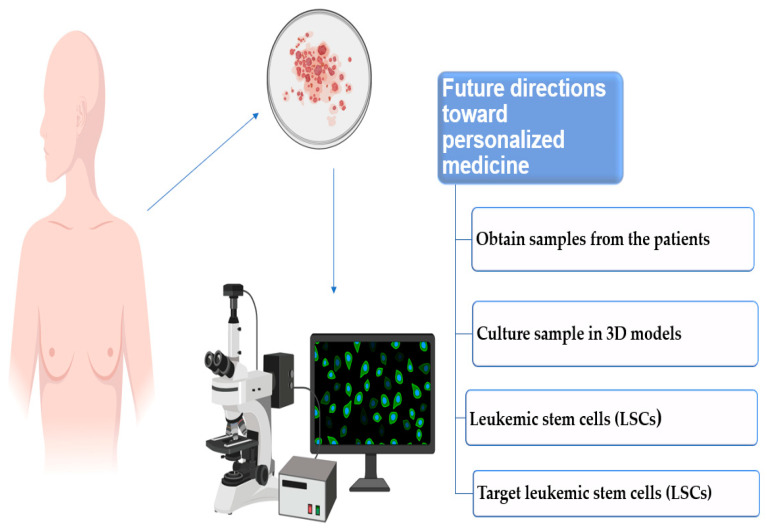
Proposed algorithm of the future utilization of in vitro 3D models in treating AML.

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
