# Peer review of "The Potential Role of 3D In Vitro Acute Myeloid Leukemia Culture Models in Understanding Drug Resistance in Leukemia Stem Cells"

_cancers, 2022, doi:10.3390/cancers14215252_

Round 1

Reviewer 1 Report

1.    This article presents an interesting preclinical review of the AML microenvironment (ME) and the in-vitro models for studying this ME. The authors will benefit from a further breakdown of the discussions/study methodology into cell-cell and cell-extracellular matrix or other pertinent soluble factors-ECM/cell interactions to enhance readability.

Still, it remains nebulous as to what takeaways the clinicians can benefit from by reading this basic science review, yet with clinical implications.  Are there any models in progress that have clinical implications, other than in-vitro studies?

2.    Suggest polishing some minor grammatical or organizational mistakes. Few examples below:

(1)    Page 2 words were missing: “….which is the notion that led to the 47 discovery of the so-called (LSC)”. What does LSC stand for? Leukemic stem cells? Also, the acronym and the full term are used loosely and interchangeably (e.g. page 7 lines 249 and 252; page 10 lines 328-332). Moreover, if the acronym  is already used once when the full term is defined, then each subsequent mention should be the acronym . Please be consistent.

(2)    Page 3 line 101: Daunorubicin. Use lower case.

(3)    Page 7 line 241: LSC and their biological behavior in vitro. This is an incomplete sentence.

Author Response

Dear Reviewer,

Thank you for reviewing our manuscript "The Potential Role of 3D In Vitro Acute Myeloid Leukemia Culture Models in Understanding Drug Resistance in Leukemia Stem Cells" to Cancers - MDPI (Manuscript ID: Cancers-1954697). We have addressed these issues in point-by-point responses below.

Point 1: This article presents an interesting preclinical review of the AML microenvironment (ME) and the in-vitro models for studying this ME. The authors will benefit from a further breakdown of the discussions/study methodology into cell-cell and cell-extracellular matrix or other pertinent soluble factors-ECM/cell interactions to enhance readability.

Still, it remains nebulous as to what takeaways the clinicians can benefit from by reading this basic science review, yet with clinical implications.  Are there any models in progress that have clinical implications, other than in-vitro studies?

Response 1:

In terms of breaking down the discussion/study methodology according to the interaction, though we could do that, the number of these models in some categories is limited and that might limit the discussion section. In terms of clinical implications, we are not aware of any of these models being used in the clinic, but we think some will be used in the future once they are tested thoroughly.

Point 2: Suggest polishing some minor grammatical or organizational mistakes. Few examples below:

(1)    Page 2 words were missing: “….which is the notion that led to the 47 discovery of the so-called (LSC)”. What does LSC stand for? Leukemic stem cells? Also, the acronym and the full term are used loosely and interchangeably (e.g. page 7 lines 249 and 252; page 10 lines 328-332). Moreover, if the acronym is already used once when the full term is defined, then each subsequent mention should be the acronym. Please be consistent.

(2)    Page 3 line 101: Daunorubicin. Use lower case.

(3)    Page 7 line 241: LSC and their biological behavior in vitro. This is an incomplete sentence.

Response 2:

  • LSC stands for Leukemic stem cells, the acronym was changed on page 7 lines 249 and 252; page 10 lines 328-332 according to the comments and uniformly changed according to section 1 in point 2
  • Daunorubicin the D was changed to a lowercase letter.
  • Page 7 line 241 the sentence was completed.

Thank you in advance for your time and effort in reviewing the resubmission of this revised manuscript.

Best regards,

Basil Al-Kaabneh, MD, Benjamin Frisch, Ph.D, Omar Aljitawi, MBBS.

Reviewer 2 Report

This review is very well written and addresses an important topic in the field. While organoids have been developed for solid cancers, 3D models for AML have not been widely used and need to be further developed. 

I have some minor points to improve the manuscript

The potential role of 3D in vitro AML culture models in understanding drug resistance in leukemia stem cells

Page 2, Line 48, please write out: leukemia stem cells (LSCs)

Page 2 Line 78: biomimicking

Page 3, Line 114: put brackets (K-562) behind AML cells

Page 4, Line 134: remove brackets 

Page 4, Line 135: remove brackets

Page 4, Line 177: remove brackets

Page 8, Line 296: BCL-2

Page 9, Figure 1 : Very poor figure. Please adjust box sizes to be equal (all different sizes) and add graphics (picture of a patient, Cell culture dish etc) to make it visually attractive.

Author Response

Dear Reviewer,

Thank you for reviewing our manuscript "The Potential Role of 3D In Vitro Acute Myeloid Leukemia Culture Models in Understanding Drug Resistance in Leukemia Stem Cells" to Cancers - MDPI (Manuscript ID: Cancers-1954697). We have addressed these issues in point-by-point responses below.

Point 1: This review is very well written and addresses an important topic in the field. While organoids have been developed for solid cancers, 3D models for AML have not been widely used and need to be further developed. 

I have some minor points to improve the manuscript

The potential role of 3D in vitro AML culture models in understanding drug resistance in leukemia stem cells

  • Page 2, Line 48, please write out: leukemia stem cells (LSCs)
  • Page 2 Line 78: biomimicking
  • Page 3, Line 114: put brackets (K-562) behind AML cells
  • Page 4, Line 134: remove brackets 
  • Page 4, Line 135: remove brackets
  • Page 4, Line 177: remove brackets
  • Page 8, Line 296: BCL-2

Response 1:

On page 2 line 48 the sentence “which is the notion that led to the 47 discovery of the so-called (LSC)”, leukemia stem cells was added.

On Page 2 line 78, biomimicing was replaced with biomimicking.

On page 3 line, 114 the bracket was moved behind AML cells

On page 4 lines 134,135 the brackets were removed from (AML-MSCs) (H-MSC) (AML-MSC), respectively.

On page 3 line 177, the brackets were removed from (Zippel, Raic).

On page 8 line 296 Bcl-2 was replaced by BCL-2.    

Point 2: Page 9, Figure 1: Very poor figure. Please adjust box sizes to be equal (all different sizes) and add graphics (picture of a patient, Cell culture dish etc) to make it visually attractive.

Response 2: On page 9 below future directions towards personalized medicine, I edited the figure and included a picture of a patient, a cell culture dish, and a microscope according to your recommendation.

Thank you in advance for your time and effort in reviewing the resubmission of this revised manuscript.

Best regards,

Basil Al-Kaabneh, MD, Benjamin Frisch, Ph.D, Omar Aljitawi, MBBS.
